

# The role of acute stress recovery in emotional resilience

Lies Notebaert[1], Roger Harris[1], Colin MacLeod[1], Monique Crane[2] and Romola S. Bucks[1,3]

[1] School of Psychological Science, University of Western Australia, Crawley, Western Australia, Australia
[2] School of Psychological Sciences, Macquarie University, Macquarie Park, New South Wales, Australia
[3] School of Population and Global Health, University of Western Australia, Crawley, Western Australia, Australia

## ABSTRACT

**Background:** Resilience refers to the process of demonstrating better outcomes than would be expected based on the adversity one experienced. Resilience is increasingly measured using a residual approach, which typically assesses adversity and mental health outcomes over a longitudinal timeframe. It remains unknown to what extent such a residual-based measurement of resilience is sensitive to variation in acute stress resilience, a candidate resilience factor.

**Methods:** Fifty-seven emerging adults enrolled in tertiary education completed measures of adversity and emotional experiences. To assess stress recovery, participants were exposed to a lab-based adverse event from which a Laboratory Stress Resilience Index was derived.

**Results:** We derived a residual-based measure of emotional resilience from regressing emotional experience scores onto adversity scores. This residual-based measure of emotional resilience predicted variance in the Laboratory Stress Resilience Index over and above that predicted by both a traditional resilience measure and the emotional experiences measure. These findings suggest that acute stress resilience may be a factor underpinning variation in emotional resilience, and that the residual-based approach to measuring resilience is sensitive to such variation in stress resilience.

## INTRODUCTION

Emotional resilience is a construct of increasing interest to businesses, organisations, public policy makers, and universities (*Anders, Frazier & Shallcross, 2012*; *Linnenluecke, 2017*; *McQuillan et al., 2021*). This interest arose from the observation that adversity is a natural part of the human experience (*Anders, Frazier & Shallcross, 2012*). Whilst it can have a profound effect on emotional well-being, there are marked individual differences in the degree to which individuals recover from the emotional impact of adversity (*Bonanno, 2004*). Emotional resilience can be defined as demonstrating more positive emotional outcomes than what would be expected based on the adversity someone experienced

Corresponding author
Lies Notebaert,
lies.notebaert@uwa.edu.au

(*Kalisch, Müller & Tüscher, 2014*). High levels of emotional resilience are associated with lower rates of mood and anxiety disorders, fewer depression symptoms following trauma exposure in childhood and adulthood, enhanced physical health and greater life satisfaction (*Joyce et al., 2018*; *Kong et al., 2015*). In contrast, poor emotional resilience is predictive of the development of anxiety and depression and increases the severity of depressive symptoms following negative life experiences (*Poole, Dobson & Pusch, 2017*; *Runkewitz, Kirchmann & Strauss, 2006*). Therefore, enhancing our understanding of why some individuals show better emotional resilience than others could be key to achieving better individual and societal outcomes.

To do so, it is critical to use a measure of emotional resilience which adequately captures individual differences in the intended concept (*Kaplan, 1999*; *Luthar, Cicchetti & Becker, 2000*; *Masten, 2007*; *Windle, Bennett & Noyes, 2011*). While the exact formulation of the concept of emotional resilience varies, nearly all formulations include reference to two key elements: exposure to adversity (ranging from daily hassles to traumatic events), and post-adversity emotional functioning (*Kalisch et al., 2017*; *Kaplan, 1999*; *Southwick et al., 2014*). Indeed, exposure to adversity is thought to be a critical condition in the measurement of resilience (*Chmitorz et al., 2018*; *Kalisch et al., 2021*). Despite this, few questionnaires designed to measure resilience adequately incorporate an assessment of exposure to adversity. *Windle, Bennett & Noyes (2011)* conducted a systematic review of resilience measurement instruments that identified 15 scales. Most focus on measuring some aspect of emotional functioning, and/or protective factors that are thought to contribute to recovery from adversity (*e.g.*, commitment, persistence, social support, personality traits). No resilience survey takes account of variation in the degree to which people have experienced adversity, such that we cannot determine whether the person being assessed recovered more readily, or less readily, than would be expected given the level of adversity experienced. The one scale that does make reference to exposure to adversity in each item is the Brief Resilience Scale (BRS, *Smith et al., 2008*), which asks respondents to indicate how well they typically recover from hard times, stressful events, or setbacks. Even in this scale, however, the level of emotional functioning cannot be ascertained relative to the adversity experienced. Thus, if two people obtain the same score on the scale but have experienced substantively different levels of adversity, we cannot conclude that they are equally resilient (*Kaplan, 2005*).

One alternative approach to measuring resilience can overcome this problem. This approach takes a measure of post-adversity emotional functioning and regresses it onto a measure of adversity experienced. The residuals obtained from this regression constitute, for each individual, the difference between their actual emotional functioning and the emotional functioning predicted by the level of adversity they experienced. In other words, this residual-based measure of resilience reflects how much better or worse someone has recovered from the emotional impact of adversity relative to what would be expected (in that sample) based on the level of adversity they experienced (*Luthar, Cicchetti & Cohen, 2006*).

This approach is increasingly used to successfully measure resilience, and to identify underlying mechanisms, risk, and protective factors (*Kalisch et al., 2021*). For example, in

an early study, *Bowes et al. (2010)* computed a measure of childhood adversity consisting of mother and child retrospective reports of bullying victimisation. Emotional problems at ages 10 and 12 were assessed by parent and teacher reports of child symptoms of withdrawal, anxiety, and depression. These measures of emotional problems were regressed on levels of bullying victimisation during primary school, and the residuals were saved as indicators of emotional resilience. These scores thus capture the degree to which children showed fewer or more than expected emotional problems over time, given their experiences of bullying victimisation. Since these early studies much research has been conducted using such residual-based measures of resilience. Many of these studies have shown that residual-based resilience measures are associated with hypothesized risk and protective factors in ways that are consistent with theory and with prior empirical findings (*Booth et al., 2020*; *Borman & Overman, 2004*; *Bowes et al., 2010*; *Kim-Cohen et al., 2004*; *Miller-Lewis et al., 2013*). One recent study showed that a residual-based measure of emotional resilience (based on adolescent exposure to adversity regressed onto adolescent psychopathology) predicted lower depressive symptoms at age 18, and significantly decreased the likelihood of participants not being in employment, education, or training at age 17 and 23 (*Cahill, Hager & Chandola, 2022*). Others have successfully used this approach to identify predictors of resilience (*Bögemann et al., 2023*; *Veer et al., 2021*) including psychosocial and genetic predictors of resilience in longitudinal designs (*Amstadter, Myers & Kendler, 2014*; *van Harmelen et al., 2017*).

The residual-based approach to measuring resilience typically assesses adversity and examines mental health outcomes over a longitudinal timeframe. That is, a residual score can be computed based on measurements covering a few months (*Kalisch et al., 2020*) to a few years (*Cahill, Hager & Chandola, 2022*). It remains unknown therefore to what extent such a residual-based measurement of resilience is sensitive to variation in acute stress resilience. The recovery of good mental health in the wake of stress exposure is theorized to be a resilience factor (*Kalisch et al., 2017*). This is widely accepted in the physiological stress literature, where a healthy hypothalamic-pituitary-adrenal (HPA) axis response to stressors involves both the successful mounting of a physiological stress response and subsequent rapid recovery. In contrast, a non-responsive HPA axis is associated with poor mental health outcomes (*Degering et al., 2023*). There is a clear temporal difference between acute stress resilience and the longer time period across which resilience is measured through the residual approach. Consequently, it is important to establish whether residual-based resilience measures are sensitive to variation in acute stress resilience. If they are, this would represent a further strength of this measurement approach.

The aim of the current study was to examine whether a residual-based measure of emotional resilience predicts acute stress resilience in the wake of a standardised adversity induced in the lab. We also examined whether the residual-based resilience measure can predict such stressor resilience over and above what can be predicted by a traditional trait-based resilience measure, and by emotional vulnerability more generally. We focused on a sample of emerging adults attending university and thus experiencing a critical transition period that can generate many positive but also stressful developmental

challenges (*Arnett, 2003*; *Sandhu, 1994*). In addition, emerging adulthood is a critical phase for establishing future trajectories in terms of educational, occupational and social attainment (*Schulenberg, Bryant & O'Malley, 2004*). Combined with the large heterogeneity in the experiences of emerging adults (*Arnett, 2007*), this developmental stage presents a pertinent sample for the investigation of emotional resilience.

## MATERIALS AND METHODS

### Participants

Participants were a convenience sample of undergraduate students at The University of Western Australia. A sample was sought with large variability in emotional experience, to enhance capacity to explain variation in emotional experience. Consequently, recruitment targeted students who varied in their disposition to experience emotional difficulties. As part of larger screening exercise, 923 students were screened using the trait subscale of the State-Trait Anxiety Inventory (STAI-T, *Spielberger et al., 1983*). Scores were divided into quartiles (by ranking them in ascending order and then splitting them into four equal parts, each containing 25% of the data), and students from each quartile were invited until 60 participants had completed the study (15 in each quartile). A sample size of 60 was deemed sufficient as it provides 0.80 power to detect a medium-to-large effect ($f^2 = 0.20$) in a linear regression analysis with two predictors. After testing finished, data from three participants whose age was above the threshold for emerging adulthood (above 30; *Arnett, 2007*) were removed. This left a final sample of 57 participants with a mean age of 19.8 (SD = 2.51), including 38 women and 19 men (none identified as non-binary)[1].

### Design

Correlation analyses were used to examine the relationship between the residual-based resilience measure and the index of lab-based stressor resilience. Hierarchical linear regression analyses tested the association between the residual-based measure of resilience and stress recovery, and examined whether the residual-based measure of resilience predicted stress recovery over and above BRS scores, and emotional vulnerability scores.

### Materials

#### Self-report questionnaires

To derive a residual-based measure of emotional resilience, a questionnaire test battery included questionnaires on past adversity and emotional experience. To compare this residual-based measure to other measures of resilience, a traditional resilience scale that references exposure to adversity was also administered.

#### Past exposure to adversity

Adversity was operationalised as the number of severe negative life events experienced in the last year, measured using the Negative Life Events Scale for Students (NLESS; *Buri et al., 2018*). The NLESS comprises 24 events relevant to student populations, selected and adjusted by five psychologist judges from 10 negative life questionnaires to cover a spectrum of severe but common adversities. Though targeted to a student sample, the scale

[1] This gender balance is representative of undergraduate psychology samples (*Gruber et al., 2021*). The design was not equally balanced for gender as the role of gender was not a focus of this study.

includes items commonly used in negative event checklists, and is therefore broadly comparable to other scales. Example events include parental divorce, having been arrested, and the death of a close friend. Respondents were asked to indicate whether or not they experienced each of these events in the past 12 months. This timeframe was selected based on the evidence in college students that negative life events experienced across the previous 12 months impact mental health symptoms (*Ji et al., 2021*). The number of 'yes' responses was summed to a total NLESS scores, with higher scores indicating greater adversity experienced. The authors have permission to use this instrument from the copyright holders.

### Past emotional experience

Emotional functioning over the past month was assessed using the 21-item version of the Depression Anxiety Stress Scale (DASS; *Lovibond & Lovibond, 1995*). The DASS was selected because of its transdiagnostic nature, assessing emotional experience across the dimensions of depression, anxiety, and stress. This is more appropriate to index emotional resilience compared to instruments that focus on symptoms in only one dimension of emotional experience (*Kalisch, Müller & Tüscher, 2014*). Respondents rate the extent to which they experienced each symptom over the past month, using a 4-point scale ranging from 0 (Did not apply to me at all), 1 (Applied to me to some degree, or some of the time), 2 (Applied to me to a considerable degree, or a good part of time), to 3 (Applied to me most of the time). There are seven items for each dimension. All items from the three scales are summed to create a DASS-21 score, with higher scores indicating more negative emotional functioning. The DASS-21 has good internal consistency reliability across both clinical and non-clinical samples (*Henry & Crawford, 2005*; *Lovibond & Lovibond, 1995*) and good convergent validity with generalised psychological distress (*Osman et al., 2012*). In the current study, the last item was not presented due to a coding error. Despite this, Cronbach's alpha for the depression (Six-items), anxiety and stress subscales demonstrated good internal consistency reliability with 0.93, 0.85 and 0.86, respectively, as did the total score (here labelled DASS-20, $\alpha = 0.95$). The DASS questionnaire is public domain.

### Traditional resilience instrument

The Brief Resilience Scale (*Smith et al., 2008*) is a six item measure which focusses on individuals' capacity to bounce back from adversity. Respondents indicate the extent to which they agree with each item on a 5-point scale from 1 (Strongly disagree) to 5 (Strongly agree). Half the items are positively worded, *e.g.*, "I tend to bounce back quickly after hard times"; half negatively worded, *e.g.*, "I have a hard time making it through stressful events". The BRS is scored by averaging item scores after reverse coding negative items. Total scores range from 1 to 5 with higher scores indicating greater resilience. Whilst the BRS has good construct validity and internal consistency ($\alpha = 0.88$ in the current study), it has only moderate content validity, test-retest reliability, interpretability, and poor criterion validity (*Windle, Bennett & Noyes, 2011*). The BRS questionnaire is public domain.

### Laboratory-based measure of emotional resilience

A laboratory-based measure of emotional resilience was used to assess acute stress resilience. This reflected individual differences in the degree to which participants recovered from the negative emotion elicited by exposure to a standardised adverse event in the lab. Participants were subjected to a standardised stressor experience (an anagram stressor task) and asked to rate their current emotional experience before and immediately after the stressor task, and then again after a 5 min recovery period. Laboratory-based emotional resilience was measured as the degree of recovery in emotional experience after exposure to this stressor, relative to the degree of negative emotion that was elicited. The task components involved in creating this measure are explained below.

### Standardised adversity delivered in lab

Adapted from *MacLeod et al. (2002)*, participants were exposed to a standardised stressor that would have a negative effect on emotional experience. The task involved solving as many anagrams as possible in 3 min, using a computer interface. Critically, many anagrams were difficult or impossible to solve, and participants received online feedback showing they were performing poorly. Participants were also videotaped and instructed that recordings of poor performance would be shown to peers in first year laboratory classes as part of course content on physiological markers of poor performance.

Participants first received instructions, after which a video camera positioned above the screen was turned on. Next, to increase anticipatory anxiety, participants were given 5 min to practice 15 solvable anagrams (fairly easy to solve) on a sheet of paper. Following this, the 3-min test period began. Letter strings were presented individually on the screen until participants had responded, or 10 s had lapsed. The letter strings were randomly selected from a battery of 40 letter strings. Twenty-six of the letter strings were solvable anagrams, as letters could be rearranged to form legitimate English words. These soluble anagrams ranged from common words, like "through" (*e.g.*, THGRUOH) to uncommon words, like "kismet" (*e.g.*, EMSTKI). The other 14 letter strings could not be rearranged to form any legitimate English word and were thus considered "insoluble anagrams", *e.g.*, "OLWGFNA". Responses were recorded by entering the solution into a textbox and pressing the 'continue' button with the mouse. Correct solutions were marked by a beep sound effect that played through the headphones. Anagrams could be skipped by clicking "continue".

To provide feedback showing that participants were performing poorly, two continuously updating performance bars were displayed on screen throughout the task. A green bar showed the rate of the participant's anagram solution, while a second (bogus) yellow bar showed the average performance of participants who had already completed the task. Additional text atop the green bar ostensibly displayed in what percentile participants were scoring relative to the average performance. The average performance bar gradually increased in size, with the participant performance bar initially increasing at the same pace and even overtaking the other bar. However, the participants' performance bar's progress

slowed, eventually falling behind the average. At the end of 3 min, all participants were shown to be performing in the bottom 10%.

When the 3 min had elapsed, the experimenter removed the video camera, instructing the participant that they were going to evaluate their performance. After 5 min the experimenter returned to provide feedback to the participant, stating that their video footage would not be used. The recording was then deleted in front of the participant.

### Emotional response to standardised lab adversity

To assess the effect of the lab-based adversity on current emotional experience, a nine-item shortened form of the DASS-21 (*i.e.*, DASS-9) was used. The DASS-9 comprised three items from each of the depression, anxiety and stress subscales that loaded highest on their respective subscale factor (*Henry & Crawford, 2005*). Respondents were asked to indicate how they feel right now, using a 4-point scale ranging from 1 (not at all) to 4 (very much). Scores on these items were summed to produce a DASS-9 score, which ranged from 9 to 36, with higher scores reflecting more negative emotional experience. The DASS-9 was administered prior to the anagram stressor task instructions being delivered (Pre-Adversity DASS-9), immediately after the 3-min anagram task had finished (Post-Adversity DASS-9), and after the feedback was given that the video recording would not be used in subsequent classes (Post-Feedback DASS-9). The internal consistency of the DASS-9 was good at each time point ($\alpha$ = 0.90, 0.89, and 0.92, respectively).

### Computation of laboratory stress resilience index

To create an index which reflected the magnitude of the dissipation in the degree of negative emotion, relative to the degree of negative emotion that was elicited by the standardised laboratory adversity, a *Laboratory Stress Resilience Index* was computed. The Laboratory Stress Resilience Index = (Post-Adversity DASS-9-Post-Feedback DASS-9)/(Post-Adversity DASS-9-Pre-Adversity DASS-9) * 100. This score expresses, as a percentage, how much DASS-9 scores dropped after the recovery period, relative to how much they increased from pre- to post-adversity. Higher positive scores indicate greater stress resilience to the lab-based adversity. One participant did not show an increase in negative emotional experience from pre to post the lab adversity, therefore the Laboratory Stress Resilience Index could not be computed for them (as the denominator was 0).

## Procedure

After being presented with the study information on screen, participants provided informed consent through mouse click before completing the questionnaires. Next, the anagram stressor was administered, including the three assessments of current emotional experience. Participants were then debriefed about the purpose of the experiment. This study protocol was approved by the Human Research Ethics office at The University of Western Australia, approval number 2021/ET000074. The data are available online: https://osf.io/wdmv8/?view_only=74e35ed653b743f688c067f9ad0434fc.

## RESULTS

### Residual-based measure of emotional resilience

On average, participants had experienced 3.68 negative life events (SD = 3.09). A residual-based measure of emotional resilience was derived from the difference between the level of emotional experience predicted by the number of past negative life events experienced, *vs* the reported level of emotional experience. Past approaches using the residual-based measure of emotional resilience have typically used the sum of the number of negative life events experienced as the measure of adversity (*Booth et al., 2020*). However, a variety of different types of negative life events exist, and they can occur in patterns when negative life events have common risk factors (*Breslau, Davis & Andreski, 1995*). Therefore, researchers have in various ways computed clusters of negative life events (*Jon, Alok & Fowler, 2015*; *Spinhoven et al., 2014*), and have shown that different clusters are differentially predictive of psychopathy (*Contractor et al., 2020*).

To determine negative life events clusters in the NLESS, a principle component analysis (PCA) was performed on the NLESS responses of an independent sample, drawn from a similar population of undergraduate students at the same university. This sample consisted of 419 participants (266 female, 148 male, and five non-binary/prefer not to say), with a mean age of 19.89 (SD = 4.58). Based on the eigenvalues and scree plot of an exploratory PCA, five components were extracted that explained 40.8% of the total variance. The items loading on each component can be seen in Table 1.

Based on these results, sum scores for each component were computed for each participant in the study, by adding up the number of negative life events they had experienced in each component. These five component life events scores controlling for age and gender were regressed on the DASS-20 scores as the outcome variable. The model coefficients are reported in Table 2.

This regression showed a significant relationship, $F(7,49) = 2.41$, $p = 0.033$, $R^2 = 0.256$, RMSE = 11.1, with a larger standardised predicted value associated with more negative emotional experience, see Fig. 1. A Residual-based Measure of Resilience was created by deriving the standardised residual from this regression for each participant. This residual reflects how much better or worse participants' emotional experience is relative to what would be predicted from the sample as a whole given that level of adversity experienced. Residuals were reverse-coded, such that greater positive residuals indicate better emotional resilience, reflecting better emotional experience than predicted by the level of adversity experienced. Greater negative residuals indicate poorer resilience, reflecting poorer emotional experience than would be predicted.

### Emotional response to the standardised lab adversity

If, overall, participants experienced the anagram stressor task as an adverse experience and recovered emotionally from this stressor after it had finished, a significant elevation in negative emotional experience should be observed from pre-adversity to post-adversity, as well as a significant decrease from post-adversity to post-feedback. To test this, DASS-9 scores were subjected to a repeated measures ANOVA with Time Point (Pre-Adversity,

**Table 1 Principle components analysis of the NLESS ($N$ = 419).** NLESS items are grouped according to the factor (PC1-PC5) for which they showed the highest loading.

| NLESS item | | Factor loading | | | | | |
|---|---|---|---|---|---|---|---|
| | | PC1 | PC2 | PC3 | PC4 | PC5 | Uniqueness |
| Factor 1: Family and personal problems | | | | | | | |
| 12 | Family has major financial pressures | 0.65 | | | | | 0.51 |
| 18 | Parents have ongoing conflicts | 0.64 | | | | | 0.50 |
| 19 | You having ongoing conflict with parents | 0.57 | 0.37 | | | | 0.53 |
| 14 | Addiction/psychological struggle of family member | 0.53 | | | | | 0.67 |
| 10 | Parent laid off work | 0.53 | | | | | 0.70 |
| 6 | Divorce of parents | 0.50 | | | | | 0.61 |
| 13 | You having major financial pressures | 0.40 | | 0.35 | | | 0.59 |
| 24 | Serious conflict with close friend | 0.35 | | | | | 0.74 |
| 17 | Serious academic problems | 0.24 | | | | | 0.88 |
| Factor 2: Abuse/assault | | | | | | | |
| 20 | You experiencing abuse/violence at home | 0.39 | 0.44 | | | | 0.54 |
| 9 | You having been assaulted | | 0.72 | | | | 0.41 |
| 22 | Unwanted sexual behaviour imposed on you | | 0.72 | | | | 0.42 |
| Factor 3: Relationships and mental health | | | | | | | |
| 16 | Cheated on by boyfriend/girlfriend | | | 0.70 | | | 0.48 |
| 11 | Serious break-up with boyfriend/girlfriend | | 0.34 | 0.57 | | | 0.56 |
| 23 | Unwanted pregnancy (either being the mother or father) | | | 0.55 | | | 0.67 |
| 15 | You struggling with addiction/psychological problem | | | 0.31 | | | 0.73 |
| Factor 4: Illness/death | | | | | | | |
| 3 | Serious illness/injury to a family member | | | | 0.66 | | 0.52 |
| 5 | Serious illness/injury to a close friend | | | | 0.55 | | 0.58 |
| 1 | Death of a family member | | | | 0.50 | | 0.70 |
| 4 | Serious illness/injury to you | | 0.33 | | 0.44 | | 0.65 |
| 2 | Death of a close friend | | | | 0.36 | | 0.81 |
| Factor 5: Natural disaster and justice system | | | | | | | |
| 21 | Family losing house through fire, flood, *etc.* | | | | | 0.72 | 0.42 |
| 8 | You arrested | | | | | 0.71 | 0.40 |
| 7 | Family member arrested | | | | | 0.57 | 0.62 |
| Eigenvalue | | 3.78 | 1.76 | 1.50 | 1.46 | 1.30 | |
| % total variance | | 16.8 | 7.3 | 6.3 | 3.1 | 5.4 | |

Note:
Varimax rotation; loadings <0.30 are not displayed except where it is the highest factor loading. The uniqueness represents the proportion of an item's variance that is not explained by the principal components.

Post-Adversity, and Post-Feedback) as the within-subjects factor. A significant main effect of Time Point was found, $F(2,112) = 44.7$, $p < 0.001$, $\eta_p^2 = 0.44$. Follow-up paired samples t-tests confirmed there was a significant increase in negative mood from Pre-Adversity ($M = 14.4$, $SE = 0.65$) to Post-Adversity ($M = 19.0$, $SE = 0.77$), $t(56) = -6.65$, $p < 0.001$, $d = 0.88$, and a significant decrease from Post-Adversity to Post-Feedback ($M = 14.0$, $SE = 0.72$), $t(56) = 8.75$, $p < 0.001$, $d = 1.16$. There was no significant difference between

**Table 2 Model coefficients of the regression predicting DASS-20 scores from age, gender, and the negative life event clusters.**

| Predictor | Estimate | SE | t | p |
|---|---|---|---|---|
| Intercept | 33.3 | 13.324 | 2.499 | 0.016 |
| Cluster 1 | 3.701 | 1.489 | 2.485 | 0.016 |
| Cluster 2 | −0.553 | 4.056 | −0.136 | 0.892 |
| Cluster 3 | 1.908 | 1.656 | 1.152 | 0.255 |
| Cluster 4 | −2.46 | 2.348 | −1.048 | 0.3 |
| Cluster 5 | −1.214 | 4.776 | −0.254 | 0.8 |
| Age | −0.842 | 0.674 | −1.25 | 0.217 |
| Gender (F-M) | −2.944 | 3.534 | −0.833 | 0.409 |

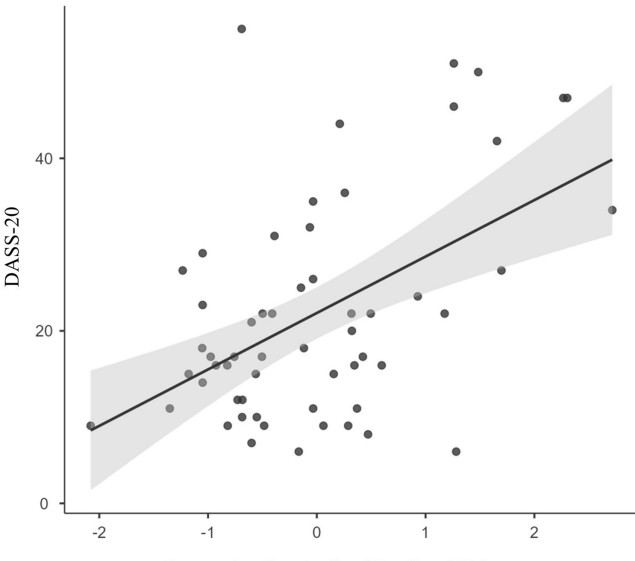

**Figure 1 The relationship between past adversity (NLESS component scores) and past emotional experience (DASS-20 scores), controlling for age and gender.** The distance from the regression line represents an individual's Residual-based Measure of Resilience.

Pre-Adversity scores and Post-Feedback scores, $t < 1$. The correlation between the Post-Adversity score and the decline from Post-Adversity scores to Post-Feedback scores was $r(55) = −0.46$, $p < 0.001$.

## Correlations between key measures

The descriptive statistics for all measures, as well as their inter-correlations, are shown in Table 3. All variables were tested for violations of normality. Skewness and kurtosis values were between −2 and +2, therefore parametric testing was deemed to be appropriate (*George & Mallery, 2010*).

The Residual-based Measure of Resilience was significantly and positively correlated with the traditional resilience scale -the BRS-(see Table 3). Higher scores on the

**Table 3 Descriptive statistics and inter-correlations of relevant questionnaires and measures.**

| Variable | Mean | SD | 1 | 2 | 3 |
|---|---|---|---|---|---|
| 1. Brief resilience scale (BRS) | 3.13 | 0.84 | | | |
| 2. DASS-20 | 22.07 | 12.93 | −0.74 | | |
| 3. Residual-based measure of resilience | 0.00 | 1.00 | 0.59 | −0.86 | |
| 4. Laboratory stress resilience index | 83.99 | 97.67 | 0.29 | −0.36 | 0.45 |

**Note:**
BRS: higher scores indicate greater self-reported emotional resilience; DASS-20: higher scores indicate greater negative affect; Residual-based Measure of Resilience, higher scores indicate better emotional resilience; Laboratory Resilience index: higher scores indicate a greater recovery in negative affect after the laboratory adversity task. *For all $|r| \geq 0.29$, $p < 0.05$. For all $|r| \geq 0.35$, $p < 0.01$.*

Residual-based Measure of Resilience (indicating greater resilience) corresponded with greater resilience as assessed using the BRS. The Residual-based Measure of Resilience was also significantly correlated, with a medium effect size and in the predicted direction, with the Laboratory Stress Resilience Index. It was correlated with a large effect size with DASS-20 scores, which is the result of the statistical properties of the regression equation used to derive the residual (*Notebaert et al., 2024*). The correlation between the traditional resilience scale (BRS) and Laboratory Stress Resilience Index was also significant and in the expected direction, though small.

## Examining the relationship between the residual-based measure of resilience and acute stress resilience

To examine the association between acute stress resilience and the residual-based measure of resilience, a hierarchical regression analysis was conducted the Laboratory Stress Resilience Index as the outcome variable. In the first step, BRS scores were included as a predictor. In the second step, the Residual-based Measure of Resilience was included as a predictor. Model comparisons were examined to determine whether the Residual-based Measure of Resilience can explain variance in recovery from a lab-based adversity over and above the variance explained by the BRS.

The first model was significant, $F(1,54) = 4.98$, $p = 0.030$, RMSE = 92.6 with 8.5% of variance in the Laboratory Stress Resilience Index scores explained by the BRS. The second step was also significant, $F(2,53) = 6.73$, $p = 0.002$, RMSE = 86.4 with 20.3% of variance in the Laboratory Stress Resilience Index scores explained by the two predictors. Model comparisons showed that the second model explained significantly more variance as compared to the first model, $F(1,53) = 7.84$, $p = 0.007$. In this second model, the Residual-based Measure of Resilience significantly predicted variation in the laboratory measure of resilience ($\beta = 0.426$, $p = 0.007$) independently from the BRS. In contrast, the BRS did not predict independent variance in the Laboratory Stress Resilience Index scores ($\beta = 0.039$, $p = 0.801$).

In a second analysis, we examined whether the residual-based measure of resilience predicts variance in recovery from lab-based adversity over and above the variance that can be predicted by overall emotional functioning over the last month. The hierarchical regression included DASS-20 scores as predictor in the first step, and added the

Residual-based Measure of Resilience in the second step. The first model was significant, $F(1,54) = 7.93$, $p = 0.007$, RMSE = 90.4 with 12.8% of variance in the Laboratory Stress Resilience Index scores explained by DASS-20 scores. The second step was also significant, $F(2,53) = 6.82$, $p = 0.002$, RMSE = 86.3 with 20.5% of variance in the Laboratory Stress Resilience Index scores explained by the two predictors. Again, model comparisons showed that the second model explained significantly more variance as compared to the first model, $F(1,53) = 5.11$, $p = 0.028$. In this second model, the Residual-based Measure of Resilience significantly predicted variation in the laboratory measure of resilience ($\beta = 0.543$, $p = 0.028$) independently from DASS-scores. In contrast, DASS scores did not predict independent variance in the Laboratory Stress Resilience Index scores ($\beta = 0.109$, $p = 0.651$).

## DISCUSSION

The aim of the current study was to examine whether a residual-based measure of emotional resilience predicts acute stress resilience in response to a standardised adversity induced in the lab. We computed a residual-based measure of emotional resilience through regressing a measure of emotional functioning onto a measure of past negative life events while controlling for age and gender. Our results showed that this residual-based measure of resilience strongly predicted emotional recovery from adversity induced in the lab. Moreover, it predicted independent variance in such a laboratory demonstration of stress resilience over and above the variance predicted by the traditional resilience scale, and over and above a measure of emotional vulnerability.

These results have two important implications: they suggest that acute stress resilience may be a factor underpinning variation in emotional resilience, and they indicate that the residual-based approach to measuring resilience is sensitive to such variation in stress resilience. Acute stress resilience in the current study was conceptualized as recovery from a laboratory-induced stressor. The anagram stress test employed as the laboratory stressor in the current study is a commonly used paradigm to examine emotional and psychophysiological responses to stress (*Notebaert et al., 2018*; *Salemink et al., 2022*; *Van Bockstaele et al., 2020*; *Watkins, Moberly & Moulds, 2008*; *Weidner et al., 2001*; *Wielgus et al., 2016*). Typically however, such research has focused on stress reactivity, whereas stress recovery has received less attention. This is consistent with the literature more broadly, which has predominantly examined the contribution of stress reactivity to resilience, while there is comparatively less research dedicated to examining the contribution of stress recovery (*Linden et al., 1997*). In a recent exception, *Degering et al. (2023)* examined whether stress reactivity and recovery as measured through cortisol responses to a standardized psychosocial laboratory stressor predicted long-term stress load and associated health effects. Results showed that cortisol stress recovery and reactivity differentially related to distinct indicators of long-term stress and downstream health consequences. Similarly, *Lee et al. (2023)* examined stress reactivity and recovery through continuous measurements of systolic and diastolic blood pressure and baroreflex sensitivity in response to a standardized laboratory protocol, and found that those with high adversity exposure showed poorer blood pressure recovery relative to a low adversity
exposure group. Building on these findings, the current study shows that acute stress recovery may be a factor contributing to increased resilience. To our knowledge, this is the first study that has demonstrated this direct relationship.

The second implication of the current study is that the residual-based approach to measuring resilience is sensitive to variation in stress resilience. The results showed that scores on a traditional resilience survey, the BRS, were also significantly associated with stress resilience, however to a smaller extent. Moreover, the Residual-based Measure of Resilience significantly predicted variation in the laboratory measure of stress resilience over and above the variance that was predicted by BRS scores. This suggests that even though the residual-based measure of resilience is derived from measures covering a much longer timeframe than the acute responses observed in response to a stress induction, they are nevertheless more sensitive to such acute stress resilience than a trait-based resilience survey.

The residual-based approach has many other advantages compared to traditional resilience measures. One is that the approach is a more direct measure of resilience itself, and not of the processes thought to underpin resilience (*Windle, Bennett & Noyes, 2011*). Although the presence of items reflecting such contributing factors (factors such as persistence or tolerance of negative affect) can be useful if the goal is to assess these process variables, it is problematic for research aiming to test hypotheses regarding the factors that may underpin individual differences in resilience. For such research, it is imperative to use a measure that is not already confounded by the inclusion of potential predictors.

Another benefit of the residual approach is that it does not exclude individuals who show negative emotional responses to adversity from being classified as resilient. This is in contrast to earlier work where only a trajectory of non-reactivity was considered a resilient response (*Galatzer-Levy, Huang & Bonanno, 2018*). By assuming a linear relationship between adversity exposure and subclinical mental health difficulties, the residual approach—in line with the notion of homeostatic resilience or resilient recovery—normalizes negative emotional responses to adversity, and only identifies a lack of resilience when these responses are disproportionate to those shown by others experiencing similar adversity (*Bonanno, 2004*; *Godara et al., 2022*; *Kalisch et al., 2017*). Normalising transient responses involving anxiety, fear, grief and depression to experiences such as threats, trauma, loss, and chronic stress can serve to prevent an escalation of the emotional response and elicit appropriate support responses from the immediate environment (*Welch, 2011*). Overall, the residual approach provides a more inclusive conceptualization of resilience by acknowledging the normality of temporary distress in the face of adversity, while still identifying disproportionate reactions as indicators of lack of resilience.

A further advantage concerns the breadth of applicability of the residual-based approach. The current study focused on emotional resilience; however, resilience can be expressed in other areas of functioning. For example, there are individual differences in students' capacity to perform academically despite originating from minority and low socioeconomic-status backgrounds (academic resilience) (*Borman & Overman, 2004*), and children vary in the degree of behavioural problems they develop when exposed to bullying

(behavioural resilience) (*Bowes et al., 2010*). There is increasing recognition that resilience is a multidimensional construct and, therefore its definition should not be restricted to specific instantiations of adversity and functioning (*Bonanno, 2004*; *Kalisch, Müller & Tüscher, 2014*; *Luthar, Cicchetti & Becker, 2000*; *Masten, 2007*; *Southwick et al., 2014*). Instead, the type of adversity and domains of functioning that are the focus of study should be conceptually driven by and tailored to the research question or applied setting. Using this residual-based approach allows the assessment of individual differences across a variety of domains of resilient functioning in a manner that takes account of variation in adversity experienced, thereby employing a consistent approach to capturing within-individual variation in different dimensions of this construct.

Of course, some limitations need to be acknowledged. The residual approach requires the assessment of multiple constructs including measures of adversity and adaptive functioning, the use of instruments that have high reliability to reduce the amount of measurement error in the variance left over from the regression model, and a large enough sample with sufficient variability to generate predicted outcome scores. Of relevance to this, in the current sample, the correlation between the Residual-based Measure of Resilience and current emotional experience scores (DASS-20) was −0.86. While a high correlation between these two variables is not unexpected, it is important to note that the Residual-based Measure of Resilience predicted variance in the Laboratory Stress Resilience Index over and above variance predicted by these DASS scores. This suggests that the Residual-based Measure of Resilience indexes a construct that cannot be captured by assessing negative emotional symptoms alone. This finding lends further weight to the strengths of this measurement approach. We also acknowledge that our DASS measure was missing one item, however due to the high internal reliability of this measure, we do not expect this missing item to have impacted the results.

Nevertheless, beyond the current proof-of concept study, one avenue for future research is to optimise this Residual-based Measure of Resilience by improving the statistical models. One way to achieve this is by increasing the variance explained in the outcome variable scores through better measurement of adversity. For example, although unavailable for the NLESS, some negative life events inventories have normative impact scores, which could be used to weight each event in the regression model (*Spurgeon, Jackson & Beach, 2001*). Doing so may lead to even more sensitive measures of resilience. In addition, retrospective reports of negative life events are subject to retrospective reporting biases (*Baldwin et al., 2019*), therefore authors may consider using prospective assessments instead. Moreover, in addition to assessing negative live events and potentially traumatic life events, future researchers may also include assessments of daily hassles. Research has shown that daily hassles have a strong relationship to mental health outcomes (*Kalisch et al., 2021*), therefore the inclusion of daily hassles assessments would render a more comprehensive picture of experiences of adversity that could impact mental health outcomes.

Further, whilst we believe that the findings are likely generalizable to other types of resilience or exposure to other types of adversities, this is yet to be established. Moreover, longitudinal data may be better able to capture dynamic recovery processes as compared to

cross-sectional data, hence this presents an exciting avenue for future research using this residual-approach (*Kalisch et al., 2020*). To overcome these limitations of having measured only one dimension of resilience at one timepoint, our study introduces a methodology which can be adopted in future studies seeking to validate alternative implementations of this residual-based measure of resilience. For example, to validate a residual-based measure of academic resilience (where the outcome is academic performance), participants could be exposed to an academic lab-based stressor (*e.g.*, a difficult performance-based task represented as an intelligence test; *Seery et al., 2013*) and the impact on academic self-esteem could be assessed (*Cassidy, 2015*).

## CONCLUSIONS

Overall, this study suggests that acute stress resilience may be a factor underpinning variation in emotional resilience, and that the residual-based approach to measuring resilience is sensitive to such variation in stress resilience. We hope that this study will support other researchers and practitioners in adopting this measurement approach, as it no doubt will greatly bolster our capacity to understand and improve resilience in all its dimensions.

### Funding
Colin MacLeod was supported by an ARC Laureate Fellowship (grant number FL170100167). The funders had no role in study design, data collection and analysis, decision to publish, or preparation of the manuscript.

### Grant Disclosures
The following grant information was disclosed by the authors:
ARC Laureate Fellowship: FL170100167.

### Competing Interests
Lies Notebaert is an Academic Editor for PeerJ.

### Author Contributions
- Lies Notebaert conceived and designed the experiments, performed the experiments, analyzed the data, prepared figures and/or tables, authored or reviewed drafts of the article, and approved the final draft.
- Roger Harris conceived and designed the experiments, performed the experiments, analyzed the data, authored or reviewed drafts of the article, and approved the final draft.
- Colin MacLeod conceived and designed the experiments, authored or reviewed drafts of the article, and approved the final draft.
- Monique Crane conceived and designed the experiments, authored or reviewed drafts of the article, and approved the final draft.
- Romola S. Bucks conceived and designed the experiments, authored or reviewed drafts of the article, and approved the final draft.

## Human Ethics

The following information was supplied relating to ethical approvals (*i.e.*, approving body and any reference numbers):

The Human Research Ethics Office at the University of Western Australia approved the study (2021/ET000074).

## Data Availability

The raw data, transformed data, and the data analysis code are available at OSF: Notebaert, Lies. 2024. "The Role of Acute Stress Recovery in Emotional Resilience." OSF. July 31. osf.io/wdmv8.

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
