# Peer review of "The role of acute stress recovery in emotional resilience"

_PeerJ, doi:10.7717/peerj.17911_

## Round 0.1 · original submission · Major Revisions

As you can see, the reviewers have offered constructive feedback that I believe will greatly assist you in revising your manuscript. I kindly request that you provide comprehensive responses to each comment from the reviewers.

·

Excellent Review

This review has been rated excellent by staff (in the top 15% of reviews)
EDITOR COMMENT
I would like to express my sincere appreciation for the reviewer's comprehensive and insightful analysis of this manuscript. The comments provided demonstrate a deep understanding of both the theoretical and experimental aspects of the study, enriched by the citation of several key studies. The constructive nature of the feedback, along with its thoroughness and amiable tone, is particularly commendable. This level of detail and consideration will undoubtedly be invaluable to the authors as they work on revising and enhancing their manuscript. Thank you for your substantial contribution to the peer review process.

Basic reporting

[note that my general remarks may be read first as an overview]
* * *
The basic reporting is generally thorough, clear and complete throughout.

My main concern is that the authors do not discuss critical papers relating to residual-based resilience quantification and arguably validating this approach. The authors cite Cahill et al. (2022) as the only example of a validation paper. However, there is a substantial body of published work that successfully uses residualised solutions to measure resilience. For example, in the Kalisch et al. (2021) paper cited in the manuscript, our research group also describes a residualised measure of resilience in adults (Kalisch et al., 2021). This approach has been successfully implemented in several publications aimed at identifying predictors of resilience (e.g. Bögemann et al., 2023; Veer et al., 2021). Other studies that have successfully used this approach include, for example, van Harmelen et al. (Van Harmelen et al., 2017) and Amstadter et al. (Amstadter et al., 2014), which identified psychosocial and genetic predictors of resilience in longitudinal designs. The above and similar literature should be cited and discussed in more detail in the manuscript.
These previous publications demonstrate the associations of established resilience-related constructs, including the BRS, with concurrent and future residualised resilience scores, thereby demonstrating the validity of the approach (although they are not explicitly framed as validation papers and do not statistically compare different approaches to resilience quantification). Given the extensive work on residual-based quantification, the authors need to correct statements about existing validation (e.g., lines 66-67, 70-73, 81) and tone down the implication of the paper regarding validation of residual-based quantification in the abstract, introduction and conclusion.

Minor points:
- Please show 95% Cis in regression (Figure 1)
- Please show the full model response of the DASS-20 scores regression by negative life event clusters. It would be very interesting to see the individual contribution of each cluster, and in particular, whether they are roughly similar in strength or diverge, where the latter would indicate that results are comparable to simply summing stressors

Experimental design

The overall experimental methodology is sound and statistical analyses are well executed.

The authors adopt an interesting approach to validate derived residualised resilience scores with an acute laboratory-based resilience measure (anagram stress task). However, and this is my second major concern, I am not convinced that this acute lab stress-recovery test is indeed a good way to validate emotional resilience. Firstly, the test itself does seem to be validated as a measure of resilience and no evidence is provided that it is a good indicator of emotional resilience (and may thus be used as a ‘ground truth’ for validation). Moreover, the lab-based resilience score operates on a very different timescale than the residualised score, which quantifies resilience relative to life event adversity exposure. Presumably, adaptation to major life stressors follows a quite different trajectory than acute resilience in the lab, with recovery over weeks and months, and may thus involve different allostatic resilience processes. Therefore, can the authors please elaborate why they believe the lab test is a good approximation to emotional resilience to major life stressors?
I would personally argue that it is extremely difficult to derive a sound alternative resilience quantification for validating the residual-based approach. The above discussed existing studies that identify longitudinal predictors of resilience may thus provide the best validation. Notably though, even if the residual-based approach is already validated, the present study indicates that acute lab-based resilience may be a particularly important predictor or residualised resilience scores, over and beyond the BRS. I would therefore really like to see a discussion of acute lab-based resilience as a potential resilience factor, and how this measure conceptually relates to residual-based resilience to more long-term adversity (i.e., life events)

- Another methodological concern relates to the selection of variables, in particular the stressor monitoring. The authors only assess LEs retrospectively over the past year. Compared to prospective assessments, such retrospective measurements are subject to retrospective reporting biases (e.g., Baldwin et al., 2019). Moreover, it has been shown that DH have a particularly strong relation to mental health outcomes (e.g., Kalisch et al., 2021) and the authors also report on the importance of smaller hassles for mental health (Lines 62-63); yet, DH are not assessed in the present study. The authors should discuss these limitations.
- The LE questionnaire used here for adversity quantification also appears to be specifically aimed at college students. From looking at the original publication (Buri et al., 2018), it appears that the questions for life events are similar to other LE questionnaires (e.g., Cochrane & Robertson, 1973) and relatively general, such that they likely apply to the broader population (of WEIRD people at least). Is that the authors assessment? This may be worth to comment on.
- Could the authors conduct an in-sample validation of stressor clusters to show that the clustering you find is not specific to your other sample? (e.g., split half and perform PCA in both – if sample size permits)

Validity of the findings

The findings are reported clearly and relevant to the study aims (with the important caveats for residual-based score validation discussed above). I find it particularly interesting to see that the residual-based measure converges better with the lab-based resilience measure than with the than with the symptom measure (DASS-20) alone, even though the DASS-20 forms part of the residual score.

The resilience field is somewhat divided on what constitutes a resilience response. Original research by Bonanno et al labels only non-reactivity as a response (Galatzer-Levy et al., 2018). More recent work proposes that both maintenance and the recovery of good mental health should be classified as resilience (Kalisch et al., 2017). Here, the authors appear to work on the assumption that only the recovery of symptoms is a resilient response. This aligns with the recovery hypothesis also common in the physiological stress literature, which suggests that a healthy HPA axis response to stressors involves both the successful mounting of a physiological stress response and subsequent rapid recovery, while a flattened HPA response indicates particularly poor mental health outcomes (e.g., Degering et al., 2023). Again, in order to judge validity of the findings, can the authors discuss more explicitly how they conceptualise the lab-based resilience measure, and carefully discuss how their results map onto different aspects of resilient responses? (i.e., would stable low stress scores during the laboratory resilience task not also indicate resilience?)
- Relatedly, I would be interested in the relationship between maximal emotional distress during the laboratory resilience task and recovery index.
- Overall, when interpreting findings, it should be made clearer that validation was conducted in a sample number of college students, using only a retrospective adversity measure of life events, and a measure of acute laboratory resilience. The authors should discuss these limitations

Additional comments

With the present study, the authors aim to validate a residual-based measure of emotional resilience, defined as emotional functioning relative to adversity exposure. They use a traditional resilience questionnaire, the brief resilience scale (BRS), and an acute lab-based resilience measure for validating the derived residualized outcome score. Results appear to suggest that the residualized score provides a better indication of resilience than the BRS and than the non-residualised measure of emotional functioning.
The authors address a question of great relevance to psychological resilience research with an interesting study design using sound methods and statistical analyses. The manuscript is well written and clearly structured. My two main concerns are that firstly, there are already several well-executed studies that arguably provide substantial validation for residual-based resilience quantification in adolescents and adults. This work needs to be cited and discussed more in the introduction and discussion. Secondly, I have concerns about the use of the acute laboratory-based resilience test for validating the residualized score. While an interesting idea, I am not convinced that the acute resilience test actually measures emotional resilience as conceptualised here. Rather, the test may be an indicator of acute stress regulation. The study results may then be interpreted as identifying good acute stress regulation as a predictor of residual-based resilience (i.e., as a resilience factor).
I believe that with the necessary changes and proper contextualisation of results, the paper will make a valuable and timely contribution towards further establishing residual-based approaches for resilience measurement and identifying potential predictors of resilience.


Further minor points
o Could the authors discuss the high correlation (r=-.86) of the residual-score with the original symptom score?
o Add to the following point to the limitations, with consideration of possible implications: In the current study, the last item was not presented due to a coding error (line 178-179)
o 328: space missing


References cited above:
- Amstadter, A. B., Myers, J. M., & Kendler, K. S. (2014). Psychiatric resilience: Longitudinal twin study. British Journal of Psychiatry, 205(4), 275–280. https://doi.org/10.1192/bjp.bp.113.130906
- Baldwin, J. R., Reuben, A., Newbury, J. B., & Danese, A. (2019). Agreement between prospective and retrospective measures of childhood maltreatment: A systematic review and meta-analysis. JAMA Psychiatry, 76(6), 584–593. https://doi.org/10.1001/jamapsychiatry.2019.0097
- Bögemann, S. A., Puhlmann, L. M. C., Wackerhagen, C., Zerban, M., Riepenhausen, A., Köber, G., Yuen, K. S. L., Pooseh, S., Marciniak, M. A., Reppmann, Z., Uściƚko, A., Weermeijer, J., Lenferink, D. B., Mituniewicz, J., Robak, N., Donner, N. C., Mestdagh, M., Verdonck, S., van Dick, R., … Kalisch, R. (2023). Psychological Resilience Factors and Their Association With Weekly Stressor Reactivity During the COVID-19 Outbreak in Europe: Prospective Longitudinal Study. JMIR Mental Health, 10, e46518. https://doi.org/10.2196/46518
- Cochrane, R., & Robertson, A. (1973). THE LIFE EVENTS INVENTORY: A MEASURE OF THE RELATIVE SEVERITY OF PSYCHO-SOCIAL STRESSORS". In Journal of Psychosomatic ICesearch (Vol. 17). Pergamon Press.
- Degering, M., Linz, R., Puhlmann, L. M. C., Singer, T., & Engert, V. (2023). Revisiting the stress recovery hypothesis: Differential associations of cortisol stress reactivity and recovery after acute psychosocial stress with markers of long-term stress and health. Brain, Behavior, and Immunity - Health, 28(January), 100598. https://doi.org/10.1016/j.bbih.2023.100598
- Galatzer-Levy, I. R., Huang, S. H., & Bonanno, G. A. (2018). Trajectories of resilience and dysfunction following potential trauma: A review and statistical evaluation. In Clinical Psychology Review (Vol. 63, pp. 41–55). Elsevier Inc. https://doi.org/10.1016/j.cpr.2018.05.008
- Kalisch, R., Baker, D. G., Basten, U., Boks, M. P., Bonanno, G. A., Brummelman, E., Chmitorz, A., Fernàndez, G., Fiebach, C. J., Galatzer-Levy, I., Geuze, E., Groppa, S., Helmreich, I., Hendler, T., Hermans, E. J., Jovanovic, T., Kubiak, T., Lieb, K., Lutz, B., … Kleim, B. (2017). The resilience framework as a strategy to combat stress-related disorders. Nature Human Behaviour, 1(11), 784–790. https://doi.org/10.1038/s41562-017-0200-8
- Kalisch, R., Köber, G., Binder, H., Ahrens, K. F., Basten, U., Chmitorz, A., Choi, K. W., Fiebach, C. J., Goldbach, N., Neumann, R. J., Kampa, M., Kollmann, B., Lieb, K., Plichta, M. M., Reif, A., Schick, A., Sebastian, A., Walter, H., Wessa, M., … Engen, H. (2021). The Frequent Stressor and Mental Health Monitoring-Paradigm: A Proposal for the Operationalization and Measurement of Resilience and the Identification of Resilience Processes in Longitudinal Observational Studies. Frontiers in Psychology, 12. https://doi.org/10.3389/fpsyg.2021.710493
- Van Harmelen, A. L., Kievit, R. A., Ioannidis, K., Neufeld, S., Jones, P. B., Bullmore, E., Dolan, R., Fonagy, P., & Goodyer, I. (2017). Adolescent friendships predict later resilient functioning across psychosocial domains in a healthy community cohort. Psychological Medicine, 47(13), 2312–2322. https://doi.org/10.1017/S0033291717000836
- Veer, I. M., Riepenhausen, A., Zerban, M., Wackerhagen, C., Puhlmann, L. M. C., Engen, H., Köber, G., Bögemann, S. A., Weermeijer, J., Uściłko, A., Mor, N., Marciniak, M. A., Askelund, A. D., Al-Kamel, A., Ayash, S., Barsuola, G., Bartkute-Norkuniene, V., Battaglia, S., Bobko, Y., … Kalisch, R. (2021). Psycho-social factors associated with mental resilience in the Corona lockdown. Translational Psychiatry, 11(1). https://doi.org/10.1038/s41398-020-01150-4

·

Basic reporting

1. Refer to Emotional resilience. Line 45-47
2. References should be referred from the most recent one. Line 60-61, 63-64, 98-100, 378-379, 413
3. Replace As such with an appropriate word. Line 112
4. Please refer to the study which shows the residual-based measure of resilience would
show high concurrent and predictive validity. Line 137-138

Experimental design

1. How many students were approached initially, of which 60 were eventually recruited?
2. What criteria were used for dividing the students into quartiles?

Validity of the findings

no comments

·

Basic reporting

In the heading Background, adjust the period at the end of the second sentence after the word ‘unvalidated’.
In the heading ‘Methods’, the methods used are hard to grasp, try to break the first sentence in smaller parts so the reader can understand easily. Correct the same in lines 31 to 33.
In the heading ‘Results’, the whole second sentence starting from ‘The residual-based measure..’ is not easy to understand and has grammatical errors as well. Correct the same in lines 36 to 38.
Check grammar in lines 76 and 77.
Language structure is not easy to understand, line 145 to 148.
Check grammar, and change the word ‘are’ to ‘were in line 161.
Check sentence structure in lines 333 to 335
Check the sentence structure in line 382
Use a supporting and relevant reference in line 46.
Use a supporting and relevant reference in line 47.
Correct the reference format in line 67.
Correct the reference format in line 90.
Correct the reference format in line 202.
Table 1 needs to be explained in more detail.

Experimental design

Why were the number of participants not evenly balanced between genders? Line 142 and 143.
Why do the data of Past Exposure to Adversity date back to 12 months while the data of Past Emotional Experience date back to 1 month? It is a possibility here that a participant was exposed to an adversity 11 months back while another was exposed 1 month back only. Line 157 and 166
Mention the list of symptoms in line 172.
Mention all the 4-point scale range in line 172 and 173.
Provide more clarity on the seven mentioned items in line 173 and 174.
Mention the 6 items in line 183.
Mention the duration of recovery period in line 198.

Validity of the findings

no comments

---

## Round 0.2 · accepted · Accept

Thank you for addressing the reviewers' concerns.

·

Basic reporting

The revisions have enhanced the clarity, robustness, and overall quality of the manuscript. All major concerns have been resolved with improved explanations. Updated and relevant references are included, with appropriate in-text citations.

Experimental design

Methodologies are now well-justified and described, ensuring scientific rigor. Data presentation is clearer, with recalculated statistical analyses verifying the accuracy of the results.

Validity of the findings

The discussion section is expanded, and the conclusions are strengthened based on the presented data.